# Multiple Endocrinology Immune-Related Adverse Events (irAEs) Related to Pembrolizumab as Neoadjuvant Treatment in Two Cases of TNBC Patients: Case Reports and Literature Review

**DOI:** 10.3390/curroncol33010028

**Published:** 2026-01-04

**Authors:** Khashayar Yazdanpanah Ardakani, Gaia Passarella, Andrea Gerardo Antonio Lania, Thoma Dario Clementi, Alessandro Fanti, Francesca Fulvia Pepe, Serena Capici, Marina Elena Cazzaniga

**Affiliations:** 1School of Medicine and Surgery, University of Milano-Bicocca, 20900 Monza, Italy; 2Phase 1 Research Center, Fondazione IRCC San Gerardo dei Tintori, 20090 Monza, Italyfrancescafulvia.pepe@irccs-sangerardo.it (F.F.P.);; 3Facoltà di Medicina e Chirurgia, Università degli Studi di Brescia, 25121 Brescia, Italy; 4Endocrinology and Diabetology Unit, IRCCS Humanitas Research Hospital, 20089 Rozzano, Italy; andrea.lania@hunimed.eu (A.G.A.L.);; 5Department of Biomedical Sciences, Humanitas University, 20072 Milan, Italy

**Keywords:** pembrolizumab, immune-related adverse events, TNBC, hypothyroidism, hypocortisolism, immunotherapy, thyroiditis, hypophysitis

## Abstract

irAEs can happen in any patient receiving ICIs, either in a metastatic or non-metastatic setting. Given their increasing application in clinical practice, understanding the underlying causes and manifestations, as well as diagnosis and management, is of great importance, making it one of the most important topics to be investigated regarding the effect on a patient’s life quality as well as quality of treatment. Here, we discuss the cases of two patients with the diagnosis of TNBC, receiving an anti-PD1 mAb, pembrolizumab, in the neoadjuvant setting presenting with dual endocrine-related irAEs. The flow of diagnosis, treatment, and patient’s consideration as well as risk factors are explained and discussed by the authors, while the management of endocrine-related adverse events is explained by two endocrinologists with expertise in management of these complications.

## 1. Introduction

In recent decades, immunotherapy has emerged as a groundbreaking advancement in cancer treatment, offering extraordinary benefits not only for patients with advanced, recurrent, or metastatic cancers, but also across broader applications.

Most of the data on endocrine immune-related adverse events (irAEs) associated with immune checkpoint inhibitors (ICIs), such as pembrolizumab—particularly in the neoadjuvant setting—derive from studies conducted in various tumor types (e.g., melanoma, NSCLC, and renal cell carcinoma) or from aggregated data in the metastatic setting, such as KEYNOTE 012 (TNBC) [1] and KEYNOTE 086 [2]. Even in the perioperative setting, as a KEYNOTE-522 trial [3], endocrine toxicities are typically reported only in summary form, without detailed information on individual cases.

Against this backdrop, and in consideration of the existing literature on pembrolizumab irAEs in solid tumors and specifically in triple-negative breast cancer, the case we present highlights two distinct endocrine toxicities occurring sequentially after pembrolizumab treatment. This case demonstrates both the spectrum of endocrine irAEs associated with ICIs and the potential reversibility of such toxicities.

## 2. Case Presentation

### 2.1. Case Number 1

A 49-year-old woman, with the diagnosis of breast cancer (BC), came to clinics for oncology consultation and initiation of treatment. Her past medical history was unremarkable, with no major illnesses or previous malignancies. Mammography and ultrasound revealed a mass in the upper outer quadrant of the left breast measuring 40 × 30 mm, along with a suspicious lymph node measuring 13 mm. On clinical examination, the size of the tumor measured 60 × 75 mm, and a mobile lymph node was palpable in the axilla.

A PET scan using Fluoro-deoxy glucose (FDG) demonstrated a high FDG uptake in the upper outer quadrant of the left breast (SUV = 28.92) and in the corresponding axillary lymph node (SUV = 9.55), with no evidence of distant metastases. Histopathological evaluation of the core biopsy sample showed invasive ductal carcinoma characterized by ER-negative, PgR-negative, HER2-0 status, and a Ki-67 index of 70%. The clinical stage was determined as cT2N+M0.

Given the diagnosis of triple-negative breast cancer (TNBC), the patient was scheduled to receive neoadjuvant therapy according to the KEYNOTE-522 regimen: four cycles of pembrolizumab, carboplatin, and paclitaxel, followed by four cycles of pembrolizumab, cyclophosphamide, and epirubicin.

By the end of the second cycle, the patient developed fever, vomiting, and marked fatigue, leading to discontinuation of carboplatin, while pembrolizumab and paclitaxel were continued. During the same treatment cycle, she developed liver dysfunction, with elevated transaminases (AST = 57 (U/L), ALT = 106 (U/L)). Consequently, the treatment regimen was modified, and paclitaxel was reduced to 85% of the standard dose.

At the completion of the fourth cycle, routine laboratory testing revealed significant hypothyroidism: thyroid-stimulating hormone (TSH) = 87.8 uUI/mL (reference range: 0.270–4.200), free-T3 (FT3) = 0.9 pg/mL (range: 2.5–4.3), and free T4 (FT4) = 2.5 pg/mL (range: 9.3–17.1) (ECLIA). Adrenocorticotropic hormone (ACTH) concentration and 8 h cortisol tests were within normal limits (ACTH0 31.4 pg/mL, 8 h cortisol = 17.3 ug/dL). Following the diagnosis of hypothyroidism, levothyroxine therapy was initiated based on the patient’s body weight of 57 Kg, starting at 75 micrograms a day and later increased to 112 micrograms per day, due to persistent thyroid insufficiency (Table 1).

The second part of the treatment was resumed without pembrolizumab administration for the first cycle, and it was reintroduced during the second cycle of epirubicin and cyclophosphamide, after consultation with the endocrinologist that confirmed the absence of contraindications.

After the second cycle, the patient was admitted to the clinic with the chief complaints of fever, fatigue, muscle pain, nausea, and loss of appetite. Blood analysis revealed markedly low levels of cortisol, 1.28 mc/dL (normal range: 5–23 mc/dL). Subsequently, cortisone acetate at a dose of 25 mg in the morning and 12.5 mg in the afternoon was prescribed. The third and fourth cycles of the treatment did not include pembrolizumab, given the limited supporting evidence in the literature for continuation under this condition.

Due to the presence of adrenal insufficiency, hypophysitis was suspected. Upon this suspicion, in November 2024, a brain MRI was performed, confirming the diagnosis of hypophysitis with normal parameters considering the size and morphology of the pituitary gland. Subsequently, proper treatment and hormone replacement were pursued. Surgery was performed, with evidence of residual disease, ypT1cypN0 (AJCC 8th edition) (17 December 2024). Histological examination revealed invasive carcinoma of no special type, triple-negative, grade 3, with areas of fibrosis and chronic inflammation consistent with neoadjuvant therapy; residual cancer burden class 2ii according to [4] (neoplastic residue 10–50%). The biological characteristics were as follows: ER (-), PgR (-), Ki67 70%, HER2 score 0. No metastases were detected in the 22 excised lymph nodes.

Given the contraindication of pembrolizumab, the patient received capecitabine for eight cycles, completed in June 2025. Table 1 and Figure 1 illustrate the longitudinal evolution of thyroid and adrenal functional parameters.

Follow-up exams in April 2025 were consistent with hypothyroidism—TSH: 38.9: UI/mL; FT3/FT4: 2.1/12 pg/mL; the values indicated coexistent hypothyroidism upon which endocrinology consultation was advised and the patient started hormone replacement with thyroid hormones after the endocrinology consult; thyroid hormone replacement is still ongoing (next follow-up: January 2026).

Given the ineffectiveness of levothyroxine in the suppression of TSH, the endocrinologist started Sodic Liotironin for the patient with the aim of achieving lower levels of TSH, while the cortone acetate dosage was not modified.

The histopathological subtype of hypophysitis presented in patients due to ICI therapy is lymphocytic hypophysitis. It is worthy to notice that a normal non-contrast MRI lacks specificity and sensitivity in diagnosing this type of inflammation of the pituitary gland, which happens due to autoimmunity. The best diagnostic imaging technique in order to make the diagnosis could be a dynamic contrast-enhanced MRI, which still does not possess full accuracy, requiring lab tests and values to confirm the hypothesis of hypophysitis [5].

### 2.2. Case Report 2

A 51-year-old female patient, after the discovery of a breast nodule, underwent a bilateral mammography with breast ultrasound that revealed a 43 × 30 mm main lesion in the upper external quadrant of the right breast and three other nodes together with a suspected 20 mm ipsilateral axillary lymph node. A breast MRI confirmed these findings. The following biopsy allowed the diagnosis of invasive non-special-type triple-negative breast carcinoma, with the characteristic lack of expression of ER (-) and PgR (-), grade 3, Ki67 70%, and HER2 1+. A whole-body 18-FDG Positron Emission Tomography (PET) excluded distant metastases and confirmed the breast lesion. The patient’s medical history included surgical removal of a perianal granuloma, age of 47, a sigmoid polyp, age of 49, and a uterine polyp at the age of 52.

Considering the extension of the disease (clinical stage cT3 N+ M0) and the tumor’s biological characteristics, the patient was eligible for a neoadjuvant treatment according to the KN-522 regimen. Due to the young age of onset of TNBC, a genetic analysis was conducted without revealing any pathogenetic variants in BRCA1 and BRCA2 genes.

After the completion of the first phase of neoadjuvant therapy (pembrolizumab, carboplatin, and paclitaxel), the bilateral mammography and breast ultrasound showed a dimensional reduction in the primary lesion (36 × 19 mm) and three other nodes.

The patient started the second part of neoadjuvant therapy with epirubicin and cyclophosphamide every three weeks combined with pembrolizumab.

After the third cycle, she reported persistent nausea and pronounced fatigue. Laboratory tests revealed suppressed levels of cortisol and ACTH (0.41 mcg/dL and 1.5 pg/mL, respectively).

After the endocrinology consult, additional tests were performed, which revealed electrolytes within the normal range, an fT3 level of 5.3 pg/mL, fT4 of 6.6 pg/mL, and TSH of 1.58 uUI/mL; Prolactin was equal to 26.1 ng/mL, slightly elevated but considered non-specific. Anti-thyroglobulin and anti-thyroperoxidase antibodies were negative. In addition, a brain MRI with contrast was performed and the pituitary gland appeared normal. A diagnosis of secondary hypocortisolism due to immunotherapy-induced hypophysitis was therefore established (Table 2).

A single dose of hydrocortisone 100 mg was promptly administered intravenously followed by replacement therapy with cortone acetate at a dose of 25 mg in the morning and 12.5 mg in the afternoon. This therapy resulted in clinical improvement, gradual resolution of nausea and asthenia with the normalization of thyroid function blood levels.

Due to significant immune-related toxicity, the systemic treatment was considered completed and the patient was admitted to undergo the surgery. New bilateral mammography and breast ultrasound were performed showing substantial stability of the known lesion in the upper external quadrant (30 × 19 mm); one of the satellite nodes found in the previous radiological examination was stable (6 mm) while the other two were no longer appreciable.

The patient underwent a right mastectomy, sentinel node biopsy, and plastic reconstruction. Histological examination showed non-special-type triple-negative invasive carcinoma, grade 2, with areas of fibrosis and chronic inflammation referable to neoadjuvant therapy and residual cancer burden 2iii according to [4] (neoplastic residue > 50%). The biological characteristics were as follows: ER (-), PgR (-), Ki67 50%, HER2 score 0. No metastases were detected in the five removed lymph nodes. The final pathological stage was ypT2 ypN0, which was defined according to AJCC 8th edition.

A new 18-FDG PET scan revealed three small areas of uptake (SUV = 13.14), likely lymph nodes, along the right internal mammary chain; given the recent surgery, these nodes appeared to have an inflammatory nature but were considered suspected.

Considering the presence of residual disease and the previous treatments with related side effects, the patient started adjuvant therapy with capecitabine.

A second follow-up 18-FDG PET scan scheduled after completing three cycles of systemic adjuvant treatment demonstrated disease progression.

Therefore, the patient was considered to receive Sacituzumab Govitecan, which was administered at 70% of the planned dose due to homozygosity for the UGT1A1 7TA allele. A complete metabolic response was achieved after five months of therapy.

Given the complete metabolic response observed on 18-FDG PET, it was decided to continue with close clinical and radiological monitoring only. Currently, eight months later, the patient remains in complete response, without further treatment. Table 2 summarizes the lab values of this patient.

In January 2025, hormone replacement for glucocorticoids was modified from cortone to hydrocortisone, and the level of ACTH and basic levels of cortisol as well as TSH, FT3, and FT4 were within the normal limit. Regarding oncological disease after achieving CMR, the patient is undergoing yearly follow-ups; the last visit was performed in July 2025, with no evidence of disease recurrence.

Limited clinical research and data regarding whether patients can follow their normal therapeutic plan after hormone replacement or not led to discontinuation of treatment in both cases, compromising the results achieved after the surgery. In case number two, the risk of recurrence—both distant and local—increases if only chemotherapy is pursued as a neoadjuvant regime, leading to the fact that the patient had a recurrence of the disease even after adjuvant cycles of capecitabine and only after treatment with Sacituzumab Govitecan—an antibody–drug conjugate—showed evidence of disease regression and complete metabolic response (CMR).

## 3. Materials and Methods

This paper presents two cases of early TNBC patients receiving neoadjuvant chemo-immunotherapy and who subsequently developed irAEs as a result of pembrolizumab administration. In both cases, informed consent was obtained, and the findings were discussed with the patients. All laboratory values and reports are derived from routine blood tests performed before each cycle of therapy, as well as from the reports of their medical oncologist.

The literature discussion is based on a systematic search of studies on the PD-1 and PD-L1 inhibitors/monoclonal antibodies and their systemic side effects, as well as on immune checkpoint inhibitors (ICIs) and associated irAEs, conducted through PubMed.

## 4. Results

### 4.1. Adverse Events of Immune Checkpoint Inhibitors

Our body’s immune system becomes activated upon encountering foreign antigen, but it can also react against self-antigens. When this occurs, the response is termed an autoimmune reaction. To prevent such harmful self-reactivity, the body has evolved several different regulatory mechanisms to inhibit such responses, including immune checkpoints. In general, immune checkpoints function by inhibiting the activation of immune cells upon antigen recognition, thereby suppressing immune responses against self-antigens and maintaining what is known as immune tolerance.

Cancer cells can exploit these mechanisms to evade immune surveillance. Consequently, inhibition of these checkpoints may enhance anti-tumor immunity. However, this therapeutic strategy may impair the evolved immune tolerance, leading to autoimmune reactions affecting various organs, known as immune-related adverse events (irAEs) [6].

### 4.2. Adverse Events of Pembrolizumab

A recent systematic review and network meta-analysis that included 15,370 patients from 36 head-to-head phase II and III randomized trials ranked the general safety of ICIs [7]. Pembrolizumab demonstrated an intermediate safety profile (probability of irAEs 55%, pool incidence 75.1%), with a comparatively low risk of endocrine toxicities. The predominant treatment-related adverse events for pembrolizumab were arthralgia, pneumonitis, and hepatic toxicities [7].

A Belgian systematic review and meta-analysis of 101 studies involving 19,922 patients (predominantly melanoma, non-small-cell lung cancer (NSCLC), and renal cell carcinoma) further estimated the incidence of endocrine irAEs. Among the 4485 treated with pembrolizumab, the reported incidence was 8.5% for hypothyroidism, hypophysitis in 1.1% of the cases, and 0.8% for adrenal insufficiency [8].

### 4.3. Pembrolizumab in Breast Cancer

Keynote-522 changed the outlook and prognosis of TNBC patients. However, revolutionary pembrolizumab is not the first ICI that was tested in these patients. Prior to pembrolizumab, the IMpassion-130 trial evaluated the role of atezolizumab—an anti-PD-L-1 monoclonal antibody—in combination with nab-paclitaxel in TNBC patients with advanced-stage disease.

IMpassion-130 reports a significant difference in the atezolizumab group than the placebo in the progression-free survival (PFS) rate—(7.2 months vs. 5.5 months; hazard ratio 0.80, CI = 0.69–0.92; *p* = 0.002)—providing evidence for survival improvement. The provided safety profile of IMpassion-130 indicates a higher number of irAEs in the atezolizumab group—259 (57.3%)—than placebo—183 (41.8%). Immune-related hypothyroidism comes in first place of irAEs in this study in both groups with a higher incidence in the atezolizumab arm {78 (17.3%) vs. 19 (4.3%)}. The second most common irAEs would be immune-related hepatitis, followed by hyperthyroidism (20), pneumonitis (14), meningoencephalitis, and colitis with equal distribution (5) [9].

A phase 2 adaptive trial evaluated the efficacy of a short course of neoadjuvant ICIs without concurrent chemotherapy in early-stage TNBC, within three cohorts of patients. Cohort A followed a treatment plan of four weeks of nivolumab; in cohort B, four weeks of nivolumab plus ipilimumab was administered and finally cohort C belonged to the patients with high TILs (≥50%) who were administered six weeks of neoadjuvant nivolumab + ipilimumab. They observed immune activation in 53% of patients of cohort A, and 60% of cohort B (cohort C’s primary endpoint did not include immune activation). Overall, 53% of patients in cohort C achieved a major pathologic response (MPR; ≤10% viable tumor), with 33% being the rate of pCR achievement.

The BELLINI trial points out two important points. Firstly, the levels of response and immune activation are directly correlated with the levels of infiltration of the CD8+ T-cells and Follicular T-cells at the tumor site, while high levels of regulatory T-cells and the greater distance between CD8+ T-cells and the tumor are inversely related to the response rate as well as immune activation. Secondly, they report a high rate of immune-related endocrinopathies (53%), with a 17% rate of grade ≥ 3 [10].

Keynote-522 reported a 13% incidence of grade ≥ 3 immune-related adverse events (irAEs) in patients receiving pembrolizumab.

A real-world multicenter study from ten cancer centers in Brazil, Neo/GBECAM 0123, evaluated the prevalence of irAEs in patients with TNBC treated with pembrolizumab. Among 368 patients, 31% experienced serious irAEs of any grade, a rate comparable to that observed in KN-522 (38.9%). Notably, irAEs occurred more frequently during the neoadjuvant course of the treatment (72.8%), compared with the adjuvant phase (27.2%).

In this study, the most common irAEs were endocrinopathies (12.8%), followed by cutaneous reactions (7.6%) and gastrointestinal toxicities (7.1%).

However, among grade ≥ 3 irAEs, gastrointestinal toxicities were the most prevalent (32%). Overall, the same safety profile and frequency of irAEs were consistent with those reported in Keynote-522 [11].

Cortes et al., in the Keynote-355 (KN-355) trial, evaluated the efficacy of pembrolizumab combined with chemotherapy versus placebo plus chemotherapy in advanced TNBC. The addition of pembrolizumab resulted in superior outcomes, with a median overall survival (MOS) of 23 months compared with 16.1 months in the placebo group [12].

Regarding safety, 26.5% of patients in the pembrolizumab arm experienced irAEs of any grade, and 5.3% developed irAEs of grade 3–4. Hypothyroidism (15.8%) and hyperthyroidism (4.3%) were the most frequently reported irAEs [12].

Table 3 summarizes the data regarding these three important trials.

The real-world data on irAEs occurring in TNBC patients are not widely available; however, Jyan et al. reported a real-world analysis of the complications in TNBC patients undergoing an identical therapeutic schedule as Keynote-522, with pembrolizumab and chemotherapy. Eighty patients (34%) developed irAEs, among which endocrinopathies (52%) and GI disturbances (23%) were the most common complications. Two patients died after developing colitis as a result of consecutive consequences. In early-stage TNBC (stages II, III), the rate of irAEs is almost identical across different studies. This can be justified by several different factors. Firstly, almost all the patients tested are women; considering that gender is an important factor involved in the predisposition of patients to autoimmunities, we might expect a higher rate of irAEs in this class of patients. This is consistent with the results reported by these studies. While in KN-522 the rate of irAEs is 38.9%, this rate drops to almost 30% in KN-158 and 22% in KN-189 including both genders in the trials [13].

An important point needs to be noticed; in spite of the fact that in most of the cases, the damage to the endocrine glands is permanent, it is easily manageable with appropriate hormone replacement. In the real-world trial mentioned previously, primary hypothyroidism and hypothyroidism after an initial period of hyperthyroidism were the most common endocrine-related AEs, followed by adrenal insufficiency secondary to hypophysitis and pituitary damage as the result of autoimmunity. However, in all the cases, it can be easily manageable if proper diagnosis by the clinical oncologist is made on time, and patients are referred to receive an appropriate endocrinology consultation. In so many cases after a contemporary period of treatment suspension, pembrolizumab can be initiated under proper monitoring measures to make sure that patients will achieve the best oncological therapy regarding the aggressive nature of their disease. Pembrolizumab administration in TNBC patients not only remarkably improves pCR achievement in early stages of the disease, but also reduces tumor burden more than a period of neoadjuvant therapy solely with chemotherapy in patients not achieving pCR, accompanied by a substantial decrease in the rate of distant metastasis—a pCR rate of 64.8% with pembrolizumab compared to 51.2% with only chemotherapy—and a rate of distant recurrence of 7.7% in the pembrolizumab arm vs. 13.1% in the chemotherapy group.

In conclusion, proper monitoring in the course of oncology therapy is crucial to catch any complications as early as possible, to be able to provide patients with proper hormone replacements in order to pursue treatment with ICIs (check management of irAEs). Additionally, as we discussed in cases number 1 and 2, both of our cases did present with typical adverse events that both Keynote-522 and real-world analysis by Jyan et al. reported—hypothyroidism and adrenal insufficiency secondary to hypophysitis.

### 4.4. Immune-Related Adverse Events of Pembrolizumab Across Different Tumor Types

Across multiple Keynote trials, pembrolizumab consistently demonstrates a comparable safety profile, with the incidence of immune-related adverse events (irAEs) remaining remarkably similar across studies. For example, KN-177 reported irAEs in 31% of patients, KN-158 observed a rate of 29.8%, and Keynote-826—evaluating pembrolizumab with chemotherapy with or without bevacizumab in cervical cancer—reported an incidence of 33.9%. This consistency underscores the predictability of pembrolizumab’s safety profile across diverse cancer types and treatment settings.

Table 4 provides a summary of these findings [3,12,14,15,16].

### 4.5. The Management of irAEs

In most cases, endocrine irAEs are mild (grade 1–2 according to the Common Terminology Criteria for Adverse Events, CTCAE), but they often cause permanent hormonal deficits requiring lifelong replacement therapy, usually without interruption of ICI treatment [17]. According to the ESE clinical practice guideline, baseline evaluation should include morning TSH, free T4, cortisol (considering glucocorticoid therapy), glucose, and electrolytes, repeated every 4–6 weeks, ideally before each treatment cycle [18].

**Thyroid dysfunction** is the most common endocrine irAE, more frequent with anti-PD-1/PD-L1 and further increased with combination therapy [19]. It may present as thyrotoxicosis evolving to permanent hypothyroidism, and several studies link its occurrence with improved survival [20,21]. The guidelines recommend TSH and fT4 screening before treatment, every 4–6 weeks during the first 6 months, then every 2–3 months, and every 6 months up to 2 years post-ICI since thyroid dysfunction can occur even after the discontinuation of immunotherapy [18,22,23]. Symptomatic management (beta-blockers for thyrotoxicosis, levothyroxine for overt hypothyroidism) is standard; high-dose glucocorticoids are rarely indicated except in severe cases [24]. For primary hypothyroidism, both ASCO and ESE recommend levothyroxine replacement when free T4 is below the reference range and TSH is >10 mIU/L. The usual starting dose is ~1.0 µg/kg/day (slightly higher suggested by ASCO), with lower doses (25–50 µg/day) for elderly patients or those with cardiovascular disease. In patients with subclinical hypothyroidism (TSH 4–10 mIU/L, normal free T4), routine treatment is not advised [18,19,20,21,22,23,24]. Treatment with ICIs should not be interrupted except for serious symptoms or ophthalmopathy [25].

**Hypophysitis** is mainly linked to CTLA-4 inhibitors, with ACTH deficiency in 95% of cases, often alongside secondary hypothyroidism (up to 85%) and secondary hypogonadism (up to 75%) [26]. ASCO recommends baseline ACTH, cortisol, TSH, fT4, electrolytes, and gonadal hormones; monitoring should continue every 4–6 weeks for 6 months, then spaced over 2–3 years [24,25,26,27]. MRI is indicated in cases of multiple deficits, headache, or visual symptoms. Glucocorticoid replacement is essential for ACTH deficiency, but high-dose steroids are reserved for mass effect or optic chiasm compression. Levothyroxine should be titrated to maintain free T4 in the upper reference range, with adrenal insufficiency excluded beforehand [18]. Adrenal insufficiency is usually permanent, whereas hypothyroidism may recover. Gonadal replacement may be considered in younger patients with persistent hypogonadism, unless contraindicated. Desmopressin is used for confirmed central diabetes insipidus. ASCO advises temporary ICI suspension until stabilization with hormone therapy [22].

Primary adrenal insufficiency (PAI) is a rare endocrine toxicity of ICI [28], almost exclusively after PD 1/PD-L1 inhibitor use [29]. It presents with high ACTH, renin, hyperkalemia, and hypotension due to combined glucocorticoid and mineralocorticoid deficiency. Cortisol should be checked every 4–6 weeks for 6 months, then every 6 months for up to 3 years [30]. Management is based on physiologic glucocorticoid (typically with hydrocortisone 15–25 mg/day in divided doses or, alternatively, cortisone acetate 20–30 mg/day) and fludrocortisone replacement (0.05–0.15 mg); high-dose steroids are reserved for adrenal crisis where intravenous hydrocortisone with fluid resuscitation is life-saving [18,19,20,21,22,23,24]. Axis recovery is extremely rare [31]. The ASCO guidelines suggest a temporary suspension of therapy until the patient has stabilized with hormone replacement therapy [22].

Finally, ICI-induced diabetes mellitus frequently presents with diabetic ketoacidosis at onset [18] and typically necessitates lifelong insulin therapy [32]. Glycemic homeostasis (fasting blood glucose, if possible, otherwise random blood glucose and HbA1C) should be assessed before starting ICI therapy and then before each course of therapy, as recommended by the AACE statement [33].

The management of ICI-DM consists primarily of managing DKA, if present, with standardized protocols for intravenous fluids and insulin.

Treatment of ICI-DM with steroids is not recommended and may even further exacerbate hyperglycemia [33,34]. Similar to type 1 diabetes, patients with ICI-induced diabetes mellitus require lifelong insulin therapy, which can be administered via multiple daily injections or continuous subcutaneous insulin infusion.

Figure 2 summarizes the information discussed on the management of these patients.

### 4.6. Factors Affecting Occurrence of irAE

Chennamadhavuni et al. (2022) [35] reviewed risk factors that increase the incidence of irAEs associated with ICIs. They categorized these factors into three subgroups: patient-related, agent-related, and tumor-related [35].

Autoimmune disorders are generally more prevalent in women than in men. According to the analysis by Ngo ST et al., not only do autoimmune diseases occur more frequently in women, but in certain regions of the world, some autoimmune conditions are almost exclusively seen in women. For example, in China, nearly all reported cases of psoriatic arthritis occur in women [36].

This observation raises an important question: are women more likely than men to develop irAEs during ICI therapy? Unfortunately, across the Keynote trials, none of the studies explicitly reported the gender distribution among patients who developed irAEs.

A meta-analysis addressing this topic demonstrated that men have a slightly lower incidence of endocrine irAEs, whereas women experience these events more frequently. However, specific toxicities such as hypophysitis were more common in men. Similarly, Sjogren syndrome and vascular or neurologic complications secondary to ICI therapy showed a male predominant pattern. In contrast, hematologic and gastrointestinal irAEs were distributed equally between men and women [37].

Lal et al. conducted a retrospective analysis based on the FDA Adverse Event Reporting System to examine cardiovascular irAEs associated with ICIs. They found that combination immunotherapy, as well as anti-PD-1 or anti-PD-L1 monotherapy, was associated with an elevated risk of myocarditis. Furthermore, combination immunotherapy carried a higher risk of myocarditis than anti-CTLA-4 or anti-PD-(L)1 monotherapy. In contrast, chemotherapy was associated with a higher risk of heart failure than any other immunotherapy modalities [38].

Age also appears to influence the development of irAEs. In a cohort study, Samani et al. reported that younger patients demonstrated a higher incidence of irAEs. Patients younger than 75 years experienced the highest rate of all-grade irAEs, and grade ≥ 3 irAEs were more common in individuals younger than 65 (18.9% vs. 11.0% in patients aged 75 or older) [39].

A systematic review by Khoja L et al. (2017) [40] examined patterns of irAEs across different tumor types. Their analysis included 48 studies evaluating anti-CTLA-4, anti-PD-1, and anti-PD-L1 agents: 26 studies on anti-CTLA-4 therapy, 17 on anti-PD-1 therapy, 2 on anti-PD-L1 therapy, and the remainder evaluating combination regimens. They concluded that grades 3–4 irAEs occur more frequently with anti-CTLA-4 monoclonal antibodies (mAbs) than with other classes. This reflects differences in the mechanism of immune inhibition, which likely contribute to distinct toxicity patterns. According to their findings, grade ≥ 3 irAEs—and all-grade colitis, cutaneous rash, and hypophysitis—occur more commonly with anti-CTLA-4 agents. In contrast, hypothyroidism, vitiligo, pneumonitis, and arthralgia are more frequently associated with anti-PD-1 therapy.

When comparing anti-PD-1 agents across tumor types—specifically NSCLC, renal cell carcinoma (RCC), and melanoma—the authors observed that melanoma patients experienced higher rates of cutaneous and gastrointestinal irAEs but lower rates of pneumonitis.

The review also highlighted several additional observations. First of all, cutaneous reactions tend to happen early in the course of treatment with both anti-CTLA-4 and anti-PD-1, and the longer the patient is on the therapy, the lower the risk of development of irAEs. Secondly, they hypothesized that lower occurrence of irAEs in relation to anti-PD-L1 agents could be due to the sparing mechanism of PD-L2, preserving normal immune function. Thirdly, they revealed that different tumor histologies have different irAE profiles when treated with anti-PD-1 agents. And last but not least is that their analysis did not show a dose-dependent relationship between the frequency and severity of the irAEs and medication [40].

This leads to a question: if patients develop irAEs with one class of ICIs, is it possible to switch to another class of ICIs or should all ICIs be avoided to prevent further toxicity?

A case report published in 2016 offers insight into this scenario. A 45-year-old man with BRAF-mutated metastatic melanoma was treated with ipilimumab, an anti-CTLA-4 antibody, but therapy was discontinued after he developed grade 3 colitis. He was subsequently switched to pembrolizumab, and over 20 months of therapy, he experienced no major toxicities, and an objective partial response was achieved [41].

This case illustrates that, in selected patients, switching from one class of ICI to another may be feasible and may not compromise treatment efficacy.

## 5. Discussion

The introduction of pembrolizumab into the treatment regimen for patients with TNBC has significantly improved the outlook for this historically challenging cancer subtype. However, the use of immune checkpoint inhibitors (ICIs) in cancer therapy is often associated with immune-related adverse events (irAEs). In many cases, including the two presented in this paper, these adverse events are manageable, allowing patients to continue receiving the therapy that offers them the greatest clinical benefit. In the majority of cases, after receiving appropriate treatment and a temporary suspension of one or two cycles, patients are able to resume their initial therapeutic plan and achieve relatively satisfactory oncologic outcomes.

Given the expanding scope of ICI indications, close and systematic monitoring of patients is essential to promptly identify and manage potentially life-threatening immune-related adverse events (irAEs). Simultaneously, timely and appropriate management of these complications is crucial to ensure that patients can resume and benefit from their primary cancer therapy.

A close collaboration with endocrinologists can significantly improve the management of irAEs. We suggest performing a baseline evaluation that should include morning TSH, free T4, cortisol (considering concomitant glucocorticoid therapy), glucose, and electrolytes, repeated every 4–6 weeks, ideally before each treatment cycle, while MRI is indicated in cases of multiple hormonal deficits, headache, or visual symptoms. This represents an important first step in understanding a patient’s susceptibility to developing these types of adverse effects. As previously mentioned, glycemic homeostasis (fasting blood glucose, if possible, otherwise random blood glucose and HbA1C) should be assessed before starting ICI therapy and then before each treatment cycle, as recommended by the AACE statement.

Principles of patient monitoring should be divided into two levels: general considerations applicable to all ICIs, and more targeted surveillance based on precision medicine —taking into account the specific ICIs used, the patient’s tumor, and patient’s sex.

## 6. Conclusions

Management and monitoring of these patients sometimes need a team including different medical specialists and other care givers to offer the best quality of life to the patients. The presence of a multidisciplinary team, including at least one endocrinologist, along with the availability of rapid access to diagnostic techniques such as pituitary MRI, can make pembrolizumab treatment feasible even in small oncology centers or those with limited case volumes.

## Figures and Tables

**Figure 1 curroncol-33-00028-f001:**
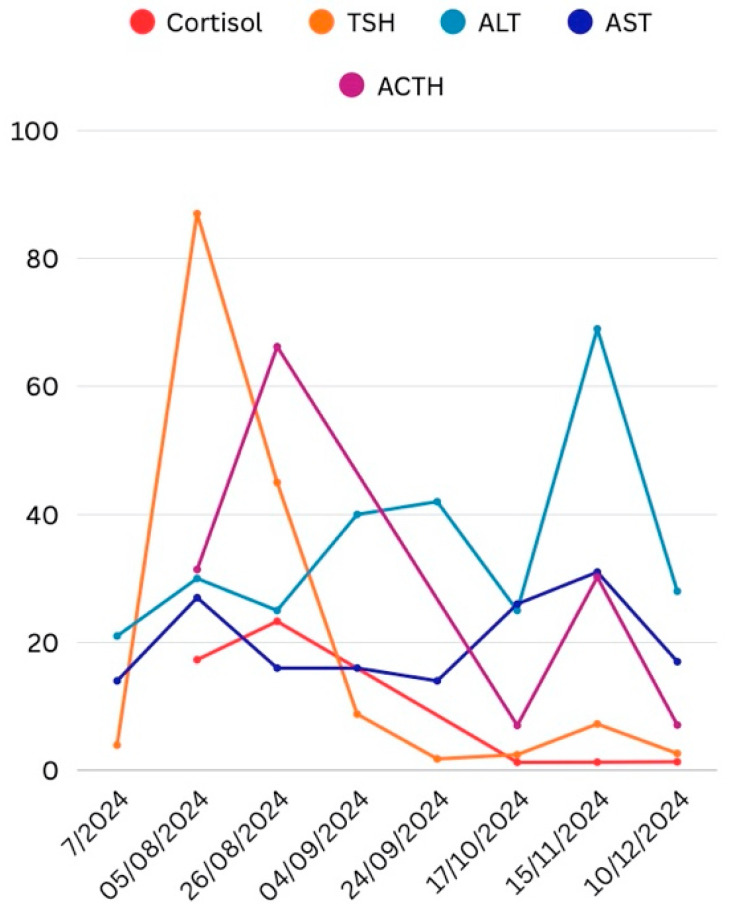
Thyroid and adrenal functional parameters.

**Figure 2 curroncol-33-00028-f002:**
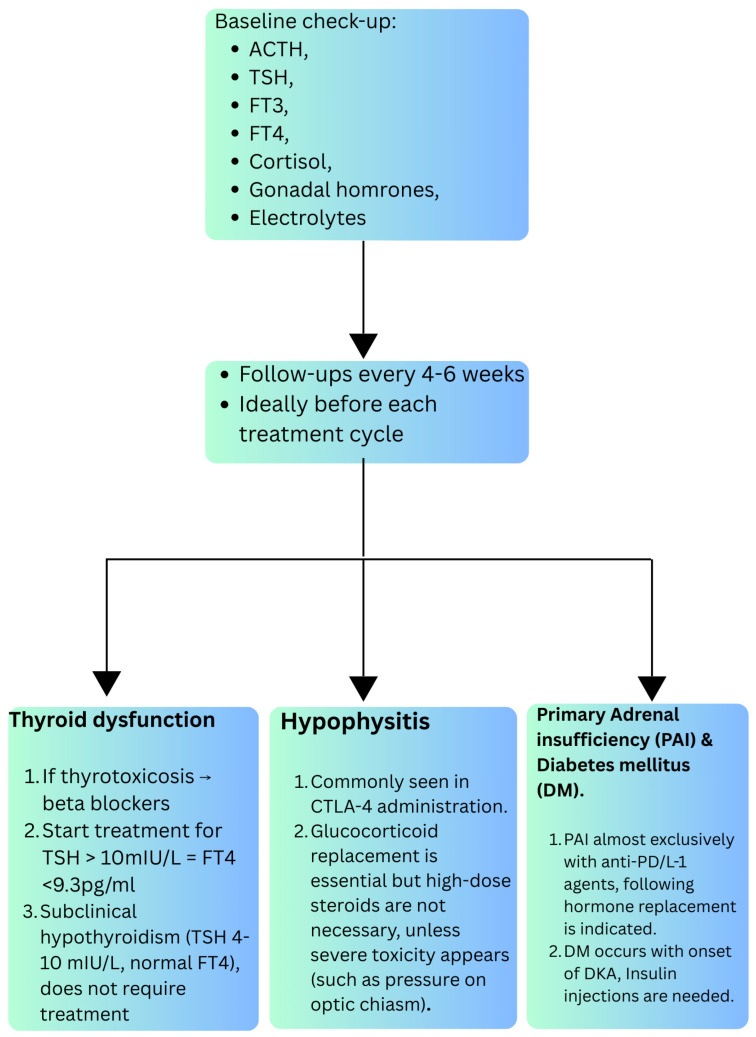
Approach to management of patients with endocrinology-related irAEs.

**Table 1 curroncol-33-00028-t001:** Lab results of patient 1.

Date	TSH (Normal Range: 0.270–4.200 uUI/mL)	FT3 (Normal Range: 2.5–4.3 pg/mL)	FT4 (9.3–17.1 pg/mL)	ACTH (Normal Range: 7.2–63.3 pg/mL)	8 h Cortisol (Normal Range: 6.24–18 ug/dL)	ALT (Normal Range: 8–41 U/L)	AST (Normal Range: 11–33 U/L)
05.08.2024	87.8	0.9	2.5			47	27
06.08.2024				31.4	17.3		
26.08.2024	45.5	1.2	8.2	66.2	23.3 (re-do on 28.08 and was 10.8)	21	16
04.09.2024	8.81	2	11.5			21	16
24.09.2024	1.81	1.8	13.3			23	14
17.10.2024	2.47	3.6	15.7	7.03	1.28	25	26
15.11.2024	7.26	2.5	12.5	30.2	1.3	69	31
10.12.2024	2.66	2.5	12.1	7.1	1.35	28	17

**Table 2 curroncol-33-00028-t002:** Lists lab values of the related blood works of patient 2.

Date	TSH (Normal Range: 0.270–4.200 uUI/mL)	FT3 (Normal Range: 2.5–4.3 pg/mL)	FT4 (9.3–17.1 pg/mL)	ACTH (Normal Range: 7.2–63.3 pg/mL)	8 h Cortisol (Normal Range: 6.24–18 ug/dL)	Prolactin (Normal Range: 4.8–23.3 ng/mL)
17.01.2023	1.58	5.3	6.6	1.5	0.41	26.1
26.01.2023	1.09	4.3	8.1	NA	NA	NA
03.02.2023	0.99	4.2	7.7	NA	NA	NA
07.03.2023	1.11	3.4	9.4	NA	0.32	NA

NA: Not available.

**Table 3 curroncol-33-00028-t003:** irAEs across most important studies on TNBC.

	KEYNOTE-522 (%)	KEYNOTE-355 (%)	IMpassion-130 (%)	BELLINI		
				Cohort A (%)	Cohort B (%)	Cohort C (%)
Overall irAEs	38.9	26.5	57.3	44	80	100
Grade ≥ 3	12.9	5.3	7.5	0	14	41
Hypothyroidism	13.7	15.8	17.3	38	47	40
Hyperthyroidism	4.6	4.3	4.4	n.r *	n.r *	n.r *
Adrenal insufficiency	2.3	n.r *	0.9	6	13	20

* n.r: not reported.

**Table 4 curroncol-33-00028-t004:** Prevalence of irAEs across various KEYNOTE studies.

	KN-522	KN-177	KN-158	KN-826	KN-355
Type of Tumor	Early TNBC	MSI-h Metastatic Colorectal Cancer	Patients with Non-Colorectal h-MSI/MMR-d Tumors (Advanced-Stage Disease)	Persistent, Recurrent, or Metastatic Adenocarcinoma, Adeno-Squamous Carcinoma, or Squamous Cell Carcinoma of Cervix	Untreated Locally Recurrent Inoperable or Metastatic TNBC
**irAEs (all grades)**	38.9%	31%	29.8%	33.9%	26.5%
**irAEs (G3 or more)**	12.9%	9%	6%	11.4%	5.3%
**Hypothyroidism**	13.7%	12%	9%	18.2%	15.8%
**Hyperthyroidism**	4.6%	4%	5.2%	7.5%	4.3%
**Hypophysitis**	n.r *	1%	n.r *	0.3%	n.r *

* n.r: not reported.

## Data Availability

The original contributions presented in this study are included in the article. Further inquiries can be directed to the corresponding author.

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
