# Peer review of "Multiple Endocrinology Immune-Related Adverse Events (irAEs) Related to Pembrolizumab as Neoadjuvant Treatment in Two Cases of TNBC Patients: Case Reports and Literature Review"

_curroncol, 2026, doi:10.3390/curroncol33010028_

Round 1
Reviewer 1 Report
Comments and Suggestions for Authors
A well researched review paper on an important topic on the importance of identifying and managing immunotherapy associated adverse effects in patients withTNBC.
Even though this paper is presented as Pembrolizumab associated immune related AE in TNBC, the paper is wide ranging with an inordinate amount of time spent discussing irAEs in other tumor types (melanoma and NSCLC). Since there is wide adoption of neoadjuvant chemoimmunotherapy with Pembrolizumab in TNBC based on the KEYNOTE-522 results, the paper is likely to have a higher impact if it is consolidated with a focus on the pattern and incidence of Pembrolizumab irAEs in TNBC.
Since this is also a literature review, the case presentations can be shorter, with elimination of unnecessary test findings such as DYPD and UGT1A1 results.
I suggest a wider vocabularly.
Author Response
"Please see the attachment."

Reviewer 2 Report
Comments and Suggestions for Authors
First, I would like to congratulate the authors for their work. The manuscript addresses a critical gap in the literature regarding endocrine irAEs in neoadjuvant pembrolizumab for TNBC, particularly the sequential development of multiple endocrine toxicities, which is understudied in early-stage disease. Their cases highlight successful collaboration between oncology and endocrinology, providing practical clinical pearls for centers managing such toxicities.
My suggestions/recommendations for strengthening and improving the manuscript, are the following:
- Strengthen Novelty: Reframe manuscript to emphasize the reversibility and non-permanent nature of sequential endocrine irAEs as key findings, distinguishing from previously reported data showing permanent deficits.
- Add Comparative Analysis: Include comparison of these two cases with KN-522 cohort irAE profiles to determine if these represent typical or atypical presentations.
- Clarify Hypophysitis Diagnosis:
- Confirm whether Case 1 hypophysitis was definitive (MRI findings described as “confirmed” but details not provided)
- Add anti-pituitary antibodies if possible; discuss role of diagnostic criteria
- Outcome Transparency:
- Update Case 2 with most recent imaging/clinical status
- Clarify ongoing hormone replacement requirements and TSH adequacy for both patients
- Discuss whether early ICI discontinuation impacted disease outcomes
- Enhanced Management Algorithm: Develop and present a clinical decision tree for: (a) timing of ICI rechallenge, (b) hormone replacement initiation, (c) multidisciplinary consultation triggers.
The manuscript has clinical merit for the Current Oncology journal audience, but requires substantive revisions to increase novelty and provide clinically actionable insights beyond standard irAE management protocols. The key contribution—sequential reversible endocrine toxicities—should be explicitly highlighted and contrasted with prior literature emphasizing permanence of these deficits.
Author Response
"Please See the attachment."

Round 2
Reviewer 1 Report
Comments and Suggestions for Authors
Important and well thought out paper on common toxicities that requires clinicians develop the ability to diagnose/identify and treat early.
The paper remains verbose and I suggest consolidating ideas, descriptions, and sentences
Author Response
Comment: Important and well thought out paper on common toxicities that requires clinicians develop the ability to diagnose/identify and treat early. The paper remains verbose and I suggest consolidating ideas, descriptions, and sentences. Reply: Appropriate changes to the figures and the manuscript had been applied to make it less verbose and more understandable, including focusing less on the side effects across other cancer types, making it briefer.